# Data-mining of potential antitubercular activities from molecular ingredients of traditional Chinese medicines

Salma Jamal, Vinod Scaria and Open Source Drug Discovery Consortium

CSIR Open Source Drug Discovery Unit, Anusandhan Bhavan, Delhi, India
GN Ramachandran Knowledge Center for Genome Informatics, CSIR Institute of Genomics and Integrative Biology (CSIR-IGIB), Delhi, India

## ABSTRACT

**Background.** Traditional Chinese medicine encompasses a well established alternate system of medicine based on a broad range of herbal formulations and is practiced extensively in the region for the treatment of a wide variety of diseases. In recent years, several reports describe in depth studies of the molecular ingredients of traditional Chinese medicines on the biological activities including anti-bacterial activities. The availability of a well-curated dataset of molecular ingredients of traditional Chinese medicines and accurate in-silico cheminformatics models for data mining for antitubercular agents and computational filters to prioritize molecules has prompted us to search for potential hits from these datasets.

**Results.** We used a consensus approach to predict molecules with potential antitubercular activities from a large dataset of molecular ingredients of traditional Chinese medicines available in the public domain. We further prioritized 160 molecules based on five computational filters (SMARTSfilter) so as to avoid potentially undesirable molecules. We further examined the molecules for permeability across Mycobacterial cell wall and for potential activities against non-replicating and drug tolerant Mycobacteria. Additional in-depth literature surveys for the reported antitubercular activities of the molecular ingredients and their sources were considered for drawing support to prioritization.

**Conclusions.** Our analysis suggests that datasets of molecular ingredients of traditional Chinese medicines offer a new opportunity to mine for potential biological activities. In this report, we suggest a proof-of-concept methodology to prioritize molecules for further experimental assays using a variety of computational tools. We also additionally suggest that a subset of prioritized molecules could be used for evaluation for tuberculosis due to their additional effect against non-replicating tuberculosis as well as the additional hepato-protection offered by the source of these ingredients.

Corresponding author
Vinod Scaria, vinods@igib.in

## INTRODUCTION

Traditional medicine still forms the mainstay of healthcare in many parts of the world. Traditional Chinese medicine (TCM) is one of the well developed and established systems of traditional medicine, and largely followed in some parts of Eastern Asia where it forms one of the major alternative medicinal practices (*Ooi, 1993*). TCM as a system of medicine was, founded almost 2,000 years ago and is dependent on the concepts of five elements and guided by the Chinese philosophy of Ying and Yang (*Qiu, 2007*; *Normile, 2003*). Recently, efforts have been underway to investigate the practice of TCM using molecular approaches. This has led to the identification and molecular characterization of ingredients used in traditional Chinese medicines (*Wang, Hao & Chen, 2007*; *Sucher, 2006*). These efforts have led to the systematic curation of the molecular structures and the biological activities of ingredients of traditional Chinese medicines (*Chen et al., 2006*; *Fang et al., 2008*; *Chen, 2011*; *Zhou, Xie & Yan, 2011*). In addition, molecular basis of the action and mechanisms of modulation (*Li & Brown, 2009*; *Wen et al., 2011*), immunomodulatory and antimicrobial activities of traditional Chinese medicines have also been actively pursued (*Nair & Abraham, 2008*; *Nader et al., 2010*).

Tuberculosis is considered one of the major tropical diseases caused by intracellular pathogen *Mycobacterium tuberculosis*. According to the World Health Organization (WHO) Global Tuberculosis Report 2012, tuberculosis causes over 1.4 million deaths annually worldwide and a major cause of morbidity and mortality especially in the developing countries in Asia and Africa (*World Health Organization, 2012*). The paucity of new drugs for the treatment of tuberculosis along with the rampant and unprecedented rise of drug-resistant strains made it imperative to discover potential new drugs for tuberculosis (*Shah et al., 2007*). The conventional process of drug discovery involves screening of large molecular libraries of molecules for biological activities, and it is a tedious, expensive and time-consuming process (*DiMasi, Hansen & Grabowski, 2003*). Data mining approaches based on cheminformatics modeling has been extensively used to prioritize molecules from large chemical datasets for specific biological activities. Such in-silico prioritization of molecules has been suggested to accelerate drug discovery by drastically reducing the time and cost-factor in conventional drug discovery processes (*Vert & Jacob, 2008*; *Melville, Burke & Hirst, 2009*; *Schierz, 2009*; *Vasanthanathan et al., 2009*).

Cheminformatics and data mining approaches have been used to mine biological activities from molecular data sets of ingredients in traditional Chinese medicines (*Li, Kong & Zhang, 2010*; *Zhang et al., 2013*). The availability of large molecular databases with systematically curated molecular data, sources and activities of ingredients of traditional Chinese medicines offer a new opportunity to use advanced data-mining tools to mine for potential activities, especially for pathogens causing neglected tropical diseases (*Chen et al., 2006*; *Fang et al., 2008*; *Chen, 2011*; *Zhou, Xie & Yan, 2011*). Previously we used high-throughput bioassay data sets to create highly accurate data-mining classifiers based on machine learning of molecular properties including antimicrobial activities for a number of neglected tropical diseases including tuberculosis, and malaria (*Periwal et al., 2011*; *Periwal, Kishtapuram & Scaria, 2012*; *Jamal, Periwal & Scaria, 2013*).

In the present report, we used one of the largest and well characterized compilation of molecular ingredients in traditional Chinese medicine and applied a host of previously generated cheminformatics models aimed at identifying potential hits with antitubercular activity against tuberculosis. We additionally employed methodologies for filtering out potential molecules using a series of in-silico filters. Our analysis revealed a total of 19 hits for antitubercular activity from the dataset. In-depth literature survey suggests 4 of these molecules are derived from plant products known to be used against tuberculosis, suggesting that the computational approach can be immensely useful in identifying and characterizing molecular activities. To the best of our knowledge, this is the first and most comprehensive data-mining and cheminformatic analysis of potential antitubercular agents from traditional Chinese medicine ingredients.

## MATERIALS AND METHODS

### Data sets

Molecular data sets of ingredients of traditional Chinese medicines were retrieved from traditional Chinese medicines Integrated Database (TCMID) (*Ruichao et al., 2013*). TCMID constitutes one of the most comprehensive online resources for ingredients used in TCM. The database hosts information on over 25,210 pure molecules retrieved from literature and other data resources.

### Computational models for antitubercular activity

The computational predictive models used in our analysis were based on the following two confirmatory screens conducted to identify novel inhibitors of *Mycobacterium tuberculosis* H37Rv, previously published by our group (*Periwal et al., 2011*; *Periwal, Kishtapuram & Scaria, 2012*). The computational models used are available online at http://vinodscaria. rnabiology.org/2C4C/models.

Briefly these models were based on two bioassays deposited in PubChem and carrying IDs AID 1332 and AID 449762. Both the assays were based on microdilution Alamar Blue assays. The former used 7H12 broth while the latter used 7H9 media. A total of 1,120 and 327,669 compounds were screened in the respective assays. The models were generated using a machine learning approach as described in *Periwal et al. (2011)* and *Periwal, Kishtapuram & Scaria (2012)*. The AID 1332 assay model was generated based on the Random forest classification algorithm and was evaluated using a variety of statistical measures which include accuracy, Balanced Classification Rate (BCR) and Area under Curve (AUC). Balanced Classification Rate is an average of sensitivity and specificity which introduces a balance in the classification rate. The model had an accuracy of 82.57%, BCR value of 82.2% and AUC value of 0.87. The AID 449762 assay model was generated based on SMO (Sequential Minimization Optimization) algorithm and was found to be 80.52% accurate, with BCR value of 66.30% and AUC as 0.75.

In addition, we created an additional model to predict the molecules active against non-replicating drug tolerant *Mycobacterium tuberculosis*. The assay was deposited in PubChem with identifier AID 488890. A total of 3,24,437 compounds were screened for

the activity. The model was generated using Random forest classification algorithm as described in the previous papers (*Periwal et al., 2011*; *Periwal, Kishtapuram & Scaria, 2012*; *Jamal, Periwal & Scaria, 2013*; *Jamal, Periwal & Scaria, 2012*) and had an accuracy of 76%, BCR value 85.2% and AUC 0.66.

## Molecular descriptors

Molecular descriptors for each of the molecules were computed using PowerMV (*Liu, Feng & Young, 2005*), popular cheminformatics software widely used to compute molecular descriptors. A total of 179 molecular descriptors were computed for each molecule. Out of the total 179 molecular descriptors, a few descriptors were pruned using bespoke scripts written in Perl depending on whether they were used in creating the respective models. We pruned a total of 29 and 25 descriptors corresponding to AID 1332 and AID 449762 respectively, while 25 were pruned for the AID 488890 model.

## Formats and format conversion

The molecules were downloaded in mol2 format and converted to SDF (Structural Data Format) format using Openbabel (*O'Boyle et al., 2011*). The molecular descriptors were converted to ARFF format compatible with Machine learning toolkit Weka (*Bouckaert et al., 2010*). We used custom scripts written in Perl for the format conversions. A complete list of scripts is also available at Crowd Computing for Cheminformatics (2C4C) repository at URL: http://vinodscaria.rnabiology.org/2C4C/models.

## SMARTS filters

The SMARTS filter is employed to remove the molecules with fragments leading to toxicity or unwanted reactivity. We used a set of SMARTS filters for the consensus candidate anti-tubercular molecules. The online server SMARTSfilter (http://pasilla.health.unm.edu/tomcat/biocomp/smartsfilter) web application was used for all comparisons. The web application was used to filter out molecules, which match to any of the five undesirable SMARTS catalogs.

### *Mycobacterium tuberculosis* permeability prediction

The small molecules could not be effective unless they are able to penetrate the cell wall. MycPermCheck (*Merget et al., 2013*) a computational tool to predict permeability of small molecules across *Mycobacterium tuberculosis*, was employed to filter the subset of potential active molecules.

## Data mining

We used Weka, a popular and freely available Data Mining Software toolkit. Predictions were performed for the dataset across the two models corresponding to assays AID 1332 and AID 449762 independently. Further, molecules predicted active in both the datasets were collated and analyzed for additional properties including activity against non-replicating drug tolerant *Mycobacterium tuberculosis* and potential to permeate the *Mycobacterium tuberculosis* cell wall. Additional filters which discount molecules with toxic

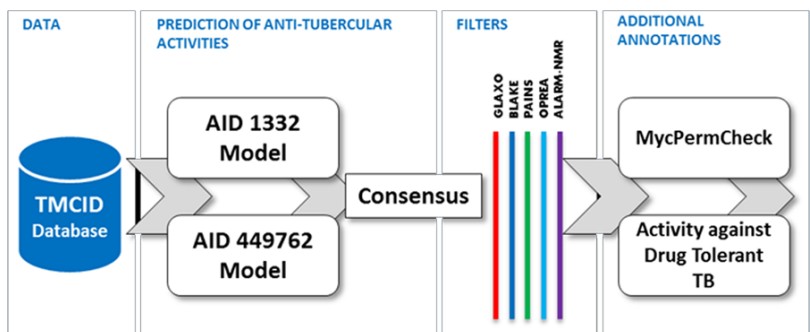

**Figure 1 Summary of the data-mining and prioritization approach involving prediction of actives, consensus building and filtering for permeability and undesirable substructures.**

fingerprints were removed using SMARTS filters. The summary of the entire workflow of prioritization is depicted as a Schema (Fig. 1).

## RESULTS

### Summary of datasets and molecules

A total of 25,210 ingredients were downloaded from Traditional Chinese Medicines Integrated Database (TCMID). We could retrieve molecular information for only 12,018 of the ingredients in the form of SMILE notations and the rest were not considered for further analysis. The molecules considered along with their SMILES are detailed in Table S1. A total of 179 descriptors were calculated using PowerMV as described above. The descriptors were further pruned for each of the models as described in the Materials and Methods section using custom scripts in Perl. This corresponds to 150 and 154 descriptors respectively for models AID 1332 and AID 449762 and 154 for AID 488890. The models, descriptors and scripts for formatting the files are available at the Crowd Computing for Cheminformatics Model Repository (http://vinodscaria.rnabiology.org/2C4C/models).

### Prediction of potential anti-tubercular hits

The 12,018 molecules obtained from TCMID were analyzed for the antitubercular activity using the computational predictive models as described above. The AID 1332 and AID 449762 models predicted 2,363 compounds and 5,864 compounds respectively as potentially active anti-tubercular. Of these molecules, a total of 1,472 molecules were predicted potential actives by both the models based on molecular descriptors and were considered for further analysis (Table S2).

Briefly we used a popular approach for filtering molecules with undesirable properties. These included briefly using SMARTS filters. Molecules which passed the filtering step were further evaluated for their effect against drug-tolerant and slow growing Mycobacterium. Molecules were further evaluated for their potential permeability with respect to the Mycobacterial cell wall.

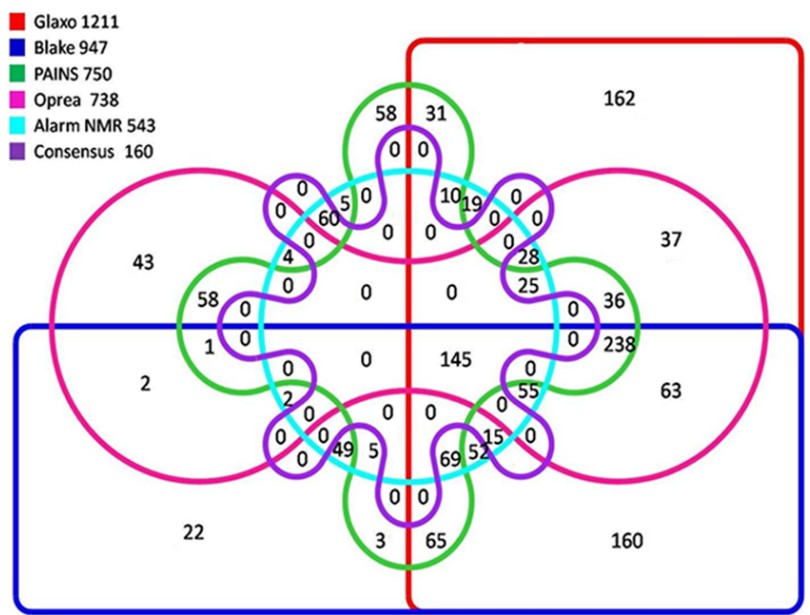

**Figure 2 Venn diagram showing active molecules filtered by any of the five SMARTS filters.**

## SMARTS filter for filtering undesirable structures

We used a set of five SMARTS filters to remove the molecules matching to any of these filters. Such substructure based filtering approach has been extensively used to prioritize molecules by filtering unwanted or potential false positives in cheminformatics screens (*Singla et al., 2013*). The SMARTS filters included 5 independent approaches namely Glaxo, PAINS, Oprea, Blake and ALARM-NMR used in tandem. Pan Assay Interference Compounds (PAINS) describes a set of substructures known to be promiscuous and have issues in high throughput assays (*Baell & Holloway, 2010*), while the Glaxo filter describes unsuitable hits or unsuitable natural products (*Hann et al., 2009*). ALARM NMR assay to detect reactive molecules by nuclear magnetic resonance (ALARM-NMR) set filters for molecules which are reactive false positives in high-throughput assays by oxidizing or alkylating a protein target (*Huth et al., 2005*). The Glaxo, Oprea and Blake filters were based on specific fitness properties. The Glaxo method involves classification of the molecules into different chemical categories based on the presence of acids, bases, electrophiles and nucleophiles in the molecule. Prior to the categorization the molecules are filtered for non-drug like properties and to remove inappropriate functional groups (unsuitable leads and unsuitable natural products) (*Hann et al., 2009*).

Out of a total of 1,472 molecules, 160 molecules passed all the filters. A total of 63.1% (929) molecules failed the ALARM NMR filter, while 49.9% (734) failed to pass Oprea filter. Similarly 49% (722) failed to pass the PAINS filter. The detailed schema showing the number of molecules failed by each filter is depicted in Fig. 2. A similar comparison of the complete set of 12,018 TCMID compounds revealed that only 1,539 compounds passed all the filters. We observed that most of the molecules did not pass through ALARM NMR

(60.7%, 7,295) molecules followed by Oprea filter (52.4%, 6,303) molecules and 5,799, 48.3% molecules could not pass through the PAINS filter.

## Molecules potentially active against non-replicating drug tolerant *Mycobacterium tuberculosis*

A total of 160 compounds filtered through SMARTSfilter were tested using a computational predictive model for potential activity against non-replicative *Mycobacterium tuberculosis*. The model predicted 19 compounds as active to act as potential inhibitors of non-replicating drug tolerant *Mycobacterium tuberculosis*. The detailed description about 19 compounds is given in Table 1. The table also shows the permeability probability of the molecules to pass through Mtb cell wall.

## Pharmacophore search in 19 molecules found to be potentially active against non-replicating drug tolerant *Mycobacterium tuberculosis*

Since pharmacophore represents the features which play a key role in the recognition of ligand by the target molecule, we generated the pharmacophore features for the 19 molecules identified to be active against non-replicating drug tolerant *Mycobacterium tuberculosis* using PharmaGist software (*Schneidman-Duhovny et al., 2008*). PharmaGist computes the pharmacophore model by doing multiple flexible alignments of the input molecules. We have reported the three highest-scoring pharmacophores which can be used for the discovery of novel drug entities (Fig. 3).

### *Mycobacterium tuberculosis* permeability prediction

We employed MycPermCheck to predict molecular permeability to Mycobacterial cell wall to estimate the potential permeability of the prioritized molecules. All the 160 molecules which passed the five SMARTSfilters were further evaluated for their ability to penetrate Mtb cell wall. Analysis revealed 9 molecules with highest probability ($>0.98$) to permeate the *Mycobacterium* cell wall barrier (Table S3).

## Literature search suggests evidence of the sources and molecules used with antitubercular properties

We further searched for the role of the plant sources of the molecules with regard to their use or known information on antibacterial or antitubercular activities. We found several molecules in herbs to have antitubercular effects. These are *Petasites japonicus* (*John, 1977*), *Piper trichostachyon* (*Wagner & Wolff, 1977*), *Solanum torvum* (*Silva et al., 2011*), *Fritillaria przewalskii* (*Chang & Paul, 2001*), *Hernandia sonora* (*Udino et al., 1999*) and *Phyllanthus urinari* (*Nair & Abraham, 2008*). In addition, many of the herbs have been shown to have hepatoprotective activities, which include *Annona reticulata* (*Thattakudian Sheik Uduman, Sundarapandian & Muthumanikkam, 2011*; *Mohamed Saleem et al., 2008*), *Annona squamosa* (*Thattakudian Sheik Uduman, Sundarapandian & Muthumanikkam, 2011*; *Mohamed Saleem et al., 2008*), and *Camellia sinensis* (*Issabeagloo & Taghizadieh, 2012*). This offers a new opportunity for new drug development considering that most of the established first-line drugs used in the treatment of tuberculosis are hepatotoxic

**Table 1** List of 19 compounds predicted as active against non replicating antibiotic tolerant *Mycobacterium tuberculosis.*

| Compound no. | Compound structure | Name | English name | Latin name | Permeability probability | Sources with antitubercular activities |
|---|---|---|---|---|---|---|
| 1. |  | F lemichapparin b | Climbing Jewelvine | Derris scandens | 0.993 | |
| 2. |  | Murrayafoline A | Taiwan Common Jasminorange, Indian Common Jasminorange, Euchretaleaf Common Jasminorange, Narrowfruit Glycosmis Root | Murraya crenulata, Murraya koenigii, Murraya euchrestifolia, Glycosmis stenocarpa | 0.98 | |
| 3. |  | 2-hexenyl benzoate | Common Tea, Szechwan Tangshen | Camellia sinensis, Codonopsis tangshen | 0.855 | |
| 4. |  | Anonaine | Hindu Lotus Large Rhizome, Bullocksheart Custardapple, Custard Apple, Chinaberry-tree Flower, Uncinate Tailgrape | Nelumbo nucifera, Annona reticulata, Annona squamosa, Melia azedarach, Artabotrys uncinatus, | 0.52 | |
| 5. |  | Orchinol | Frog Orchid, European Gymnadenia, Liriop Equivalent plant: Liriope spicata var prolifera | Coeloglossum viride [Syn. Coeloglossum viride var. bracteatum], Gymnadenia albida, Ophiopogon japonicus | 0.407 | |
| 6. |  | 1-phenyl-1-pentanone | Chuanxiong rhizome, Szechuan lovage root, Chuanxiong (Wallich Ligusticum) Equivalent plant: Cnidium officinale | Radix chuanxiong; G Rhizoma Chuanxiong, Ligusticum chuanxiong | 0.338 | |
| 7. |  | Brassilexin | India Mustard | Brassica juncea | 0.295 | |

Table 1 (*continued*)

| Compound no. | Compound structure | Name | English name | Latin name | Permeability probability | Sources with antitubercular activities |
|---|---|---|---|---|---|---|
| 8. |  | Bisacumol | Zedoary Turmeric Equivalent plant: Curcuma kwangsiensis, Common Turmeric Equivalent plant: Curcuma aromatica | Curcuma zedoaria, Curcuma longa | 0.104 | |
| 9. |  | Totarol | Longleaf Podocarpus Leaf Equivalent plant: Podocarpus macrophyllus var maki, Water Nightshade | Podocarpus macrophyllus, Solanum torvum | 0.037 | Solanum torvum |
| 10. |  | Cyclostachine a | Hairspike Pepper | Piper trichostachyon | 0.029 | Piper tri-chostachyon |
| 11. |  | Isolobinine | Indian Tobacco, Chinese Lobelia | Lobelia inflata, Lobelia chinensis | 0.018 | |
| 12. |  | Urinatetralin | Common Leafflower | Phyllanthus urinaria | 0.012 | Phyllanthus urinaria (*Nair & Abraham, 2008*) |
| 13. |  | 2-methoxy-1h- pyrrole | | | 0.004 | |

Table 1 (*continued*)

| Compound no. | Compound structure | Name | English name | Latin name | Permeability probability | Sources with antitubercular activities |
|---|---|---|---|---|---|---|
| 14. |  | Gmelofuran | Medicinal Breynia Leaf | Breynia officinalis | 0.00 | |
| 15. |  | Petasalbin methyl ether | Japanese Butterbur | Petasites japonicus | 0.00 | Petasites japonicus |
| 16. |  | Verruculotoxin | | | 0.00 | |
| 17. |  | Hinokiol | Yellowish Rabdosia | Isodon flavidus | 0.00 | |
| 18. |  | Thymine | Przewalsk Fritillary, Anhui Fritillary, Ussuri Fritillary | Fritillaria przewalskii, Fritillaria anhuiensis, Fritillaria ussuriensis | 0.00 | Fritillaria przewalskii (*Chang & Paul, 2001*) |
| 19. |  | n-methylcorydaldine | Fendler's Meadowrue, Bracteate Poppy, Asiatic Moonseed Root, Lotusleaftung | Thalictrum fendleri, Papaver bracteatum, Menispermum dauricum, Hernandia sonora | 0.00 | Hernandia sonora |

(*Yew & Leung, 2006*; *Liu et al., 2008*). We also found these molecules, Hinokiol (*Chen et al., 2009*), Totarol (*Jaiswal et al., 2007*), Murrayafoline A (*Choi et al., 2006*) and 2-hexenyl benzoate (*Yantao, Zeng & Zhang*) have been known to show antitubercular effects. The 9 molecules, which were identified by MycPermCheck, are able to penetrate the cell wall of Mycobacterium tuberculosis (Table S3). We tried to find if any of these molecules have been observed to show antitubercular or antimicrobial activity and we found 6 molecules that include Thalfinine (*Zhou, Xie & Yan, 2011*), Azulene (*Kurti & Uldrich, 1958*), Erypoegin E (*Sato et al., 2006*), Pongapinone A (*Rani et al., 2013*), Achilleol C (*Iinuma et al., 1996*) and Murrayafoline A (*Choi et al., 2006*).

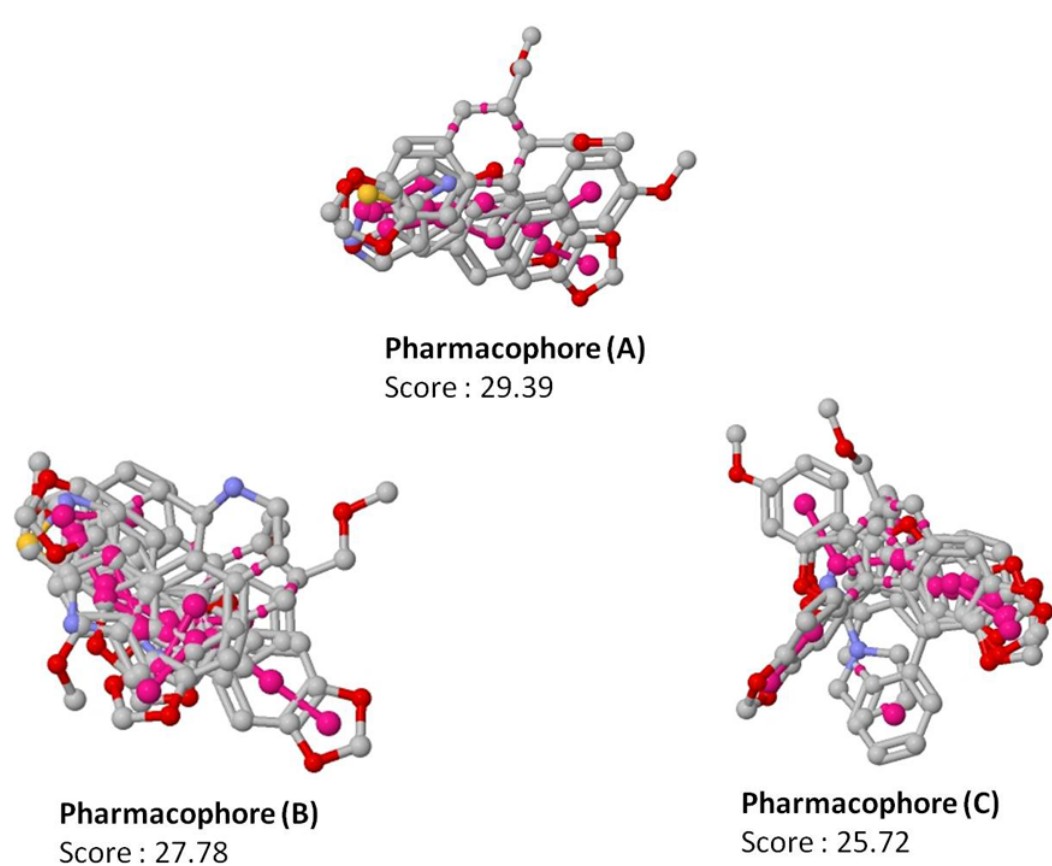

**Figure 3  Top scoring pharmacophore models (A, B and C) identified along with the alignment with the input molecules.** The pharmacophores are coloured in magenta.

## DISCUSSION AND CONCLUSIONS

Traditional Chinese Medicine (TCM) has been a major alternative medicine practice, widely followed in many parts of China and Southeast Asia (*Ooi, 1993*). Enormous efforts in the recent years have been invested in the systematic identification and characterization of the molecular activities of the ingredients and scientific validation of their effects (*Li & Brown, 2009*; *Wen et al., 2011*). The availability of well curated databases of ingredients of traditional Chinese medicines has opened up new avenues for molecular screening as well as in-silico studies, including target-based docking (*Chen et al., 2006*; *Fang et al., 2008*; *Chen, 2011*; *Zhou, Xie & Yan, 2011*). In depth screens of Chinese medicine derived compounds have been performed for a variety of pathophysiologies, including cancer (*Hu et al., 2013*), inflammatory diseases (*Han & Guo, 2012*; *Su & Hsieh, 2011*), cardiovascular diseases (*Wang et al., 2013*) and infections (*Jiang, Deng & Wu, 2013*), just to name a few. These databases are being extensively used for therapeutic development (*Cheng et al., 2010*).

Our group has used a machine learning based approach on publicly available high-throughput screen datasets to create highly accurate models for predicting specific

molecular activities against pathogens causing tuberculosis (*Periwal et al., 2011*; *Periwal, Kishtapuram & Scaria, 2012*) and malaria (*Jamal, Periwal & Scaria, 2013*). Such accurate in-silico models offer a new opportunity to prioritize large molecular databases in silico, significantly reducing the failures, cost and effort. The availability of a well-curated database of molecular ingredients of traditional Chinese medicines offer a new opportunity to mine potential active antitubercular agents and prioritize them for screening and in-depth functional assays.

In the present study, we have used two computational models based on high throughput assays on *Mycobacterium tuberculosis*. In addition to the predictive models, we used a filter based approach to filter out potential false positives/toxic molecules. Our analysis revealed a total of 1,472 molecules predicted active by both the models, of which 160 molecules passed all the five filters. These molecules were further evaluated for their permeability to mycobacterial cell wall and potential additional activity on drug-tolerant and non-replicating *Mycobacterium tuberculosis*. We also further show evidence from literature that these molecules or their sources have been used in the treatment of therapeutics. This study is not without caveats; the primary one being that the consensus approach used in the present study could be over-stringent so as to miss out on potential antitubercular hits from the screening approach. The second, being that the findings would require re-screening and in-depth functional analysis. Nevertheless we show from independent evidence that molecular ingredients or sources of the prioritized molecules have been extensively used as antibacterial or specifically in the treatment of tuberculosis. In the present study we show a proof-of-concept that data-mining approaches using accurate cheminformatics models could possibly be used to mine large datasets and prioritize molecules for antitubercular screening.

Our analysis suggests that molecular ingredients of traditional Chinese medicines offer an attractive starting point to mine for potential antitubercular agents. Chinese medicines alone (*Lu et al., 2013*) or in combination (*Li et al., 1984*) with western medicine have been explored for the treatment of tuberculosis. Potential use of Chinese medicines in combination with the standard antitubercular drugs could be an attractive alternative that could be explored in much detail. There is ample evidence in published literature that some of the ingredients of the short-listed antitubercular molecules have additional hepatoprotective action, which could be effectively used in the background of hepatotoxicity induced by the first line of drugs. We also suggest that 19 of the prioritized molecules have additional activity against drug-tolerant and non-replicating *Mycobacterium tuberculosis* suggesting that they could be potentially developed into leads for Multidrug resistant and latent tuberculosis. We hope that this report would accelerate in-depth analysis and discovery of anti-tubercular agents from molecular ingredients of traditional Chinese medicines.

## ACKNOWLEDGEMENTS

The authors thank Ms Vinita Periwal for sharing the models for antitubercular activities and for discussions. Authors also thank Dr S Ramachandran and Dr Sridhar Sivasubbu

for reviewing the manuscript and suggestions. The help and support from the National Knowledge Network (NKN) and the CDAC-Garuda grid for the connectivity and access to the computer facility is acknowledged. The Open Source Drug Discovery Consortium members are the registered members on http://sysborg2.osdd.net.

### Funding

This work was funded by the Council of Scientific and Industrial Research (CSIR), India through Open Source Drug Discovery Project (HCP001). The funders had no role in study design, data collection and analysis, decision to publish, or preparation of the manuscript.

### Grant Disclosures

The following grant information was disclosed by the authors:
Open Source Drug Discovery Project: HCP001.

### Competing Interests

Vinod Scaria is an Academic Editor for PeerJ. The authors declare there are no competing interests.

### Author Contributions

- Salma Jamal performed the experiments, analyzed the data, contributed reagents/materials/analysis tools, wrote the paper, prepared figures and/or tables, reviewed drafts of the paper.
- conceived and designed the experiments, wrote the paper, reviewed drafts of the paper.

### Data Deposition

The following information was supplied regarding the deposition of related data:
    http://vinodscaria.rnabiology.org/2C4C/models.

### Supplemental Information

Supplemental information for this article can be found online at http://dx.doi.org/10.7717/peerj.476.

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
