# Peer review of "Data-mining of potential antitubercular activities from molecular ingredients of traditional Chinese medicines"

_PeerJ, doi:10.7717/peerj.476_

## Round 0.1 · original submission · Major Revisions

This is an intetesting contribution describing data mining for traditional Chinese medication. In order to improve quality of this work authors may wish to address points raised by Reviewers 1 and 2.

Reviewer 1 ·

Basic reporting

This is an interesting paper which reports on the data mining exercise using traditional Chinese medication compounds with potential antitubercular activity. While the idea is sound and the paper should eventually be published I'd like to raise a few issues which in my opinion are of importance in this context. First, the concept behind TCM is vastly different from Western pharmacology. TCM is based on synergistic action of many components, some of which are active, some supportive and some benign. However, all of them are used in low concentrations so the action of any individual ingredient does not overwhelm the totality of the composition. For this reason I'm skeptical that searching for individual chemical entities with potency which is sufficient to act as a single drug will be successful.

Experimental design

The authors relied entirely on bioinformatic tools and no experimental validation of these findings is provided. This is still potentially useful but I would at least expect a deeper analysis in terms of common chemical features of the top ranking hits and perhaps docking to putative targets to predict their binding affinity. Ideally, a pharmacophore could be proposed by the authors which could lead to the design of novel drug entities.

Validity of the findings

It's hard to comment on the validity of the findings since these are only bioinformatic predictions without experimental validation. It would be useful for those who could validate these findings to have information about the commercial availability of the top ranking compounds.

Additional comments

some of the websites cited in the paper are hard to access or to use.

Reviewer 2 ·

Basic reporting

This is an interesting paper which offers a comprehensive approach to data mining bio-active compounds from traditional Chinese medicines. The article is clearly laid out and supporting data is comprehensive. The weak point I identify is lack of discussion and correlation of literature data on the lead molecules identified in Tables 2 and 3. Such a correlation could confirm activity of the prioritised compounds, or identify research opportunities.

Experimental design

Satisfactory

Validity of the findings

Again, I think this article could be strengthened by correlating the results of the data mining work with literature reports on bio-activity of the metabolites and TCM medicines with potential anti-tubercular activity.

Additional comments

I feel that the authors should make some amendments to the article by including some correlation with the literature, citing any documented anti-tubercular or anti-microbial activity associated with the highlighted metabolites or their originating TCM products. This could be added toTable 2 or 3.

Reviewer 3 ·

Basic reporting

The paper entitled "Data-mining of potential antitubercular activities from molecular ingredients of Traditional Chinese Medicines" is of great interest to the entire scientific community with great relevance and discoveries. This is a well written manuscript, relevant literature is appropriately referenced, the final conclusion is appropriately connected to the original question investigated. Figures are relevant to the content of the article, and appropriately described.

Experimental design

Experimental design is clearly described, methods are described with sufficient information, with a high technical standard. In the present study Authors have used two computational models based on high throughpout assays on Mycobacterium tuberculosis. Additionally, they used a filter based approach to filter out potential false positive/toxic molecules.

Validity of the findings

Data analysis technique is appropriate; final conclusions are supported by the results and appropriately stated.

Additional comments

My only suggestion is that:
Improve quality/ resolution of presented figures and compound structures in Table 1.

---

## Round 0.2 · accepted · Accept

Thank you for your submission to PeerJ. Your manuscript is Accepted for publication.